# PremPRI: Predicting the Effects of Missense Mutations on Protein–RNA Interactions

**DOI:** 10.3390/ijms21155560

**Published:** 2020-08-03

**Authors:** Ning Zhang, Haoyu Lu, Yuting Chen, Zefeng Zhu, Qing Yang, Shuqin Wang, Minghui Li

**Affiliations:** Center for Systems Biology, Department of Bioinformatics, School of Biology and Basic Medical Sciences, Soochow University, Suzhou 215123, China; 20174221042@stu.suda.edu.cn (N.Z.); 20184212006@stu.suda.edu.cn (H.L.); 20184221042@stu.suda.edu.cn (Y.C.); 1730416009@stu.suda.edu.cn (Z.Z.); 1630416001@stu.suda.edu.cn (Q.Y.); 20174221003@stu.suda.edu.cn (S.W.)

**Keywords:** Mutation, Protein–RNA interaction, binding affinity change, computational approach

## Abstract

Protein–RNA interactions are crucial for many cellular processes, such as protein synthesis and regulation of gene expression. Missense mutations that alter protein–RNA interaction may contribute to the pathogenesis of many diseases. Here, we introduce a new computational method PremPRI, which predicts the effects of single mutations occurring in RNA binding proteins on the protein–RNA interactions by calculating the binding affinity changes quantitatively. The multiple linear regression scoring function of PremPRI is composed of three sequence- and eight structure-based features, and is parameterized on 248 mutations from 50 protein–RNA complexes. Our model shows a good agreement between calculated and experimental values of binding affinity changes with a Pearson correlation coefficient of 0.72 and the corresponding root-mean-square error of 0.76 kcal·mol^−1^, outperforming three other available methods. PremPRI can be used for finding functionally important variants, understanding the molecular mechanisms, and designing new protein–RNA interaction inhibitors.

## 1. Introduction

The interactions between protein and RNA are crucial for many cellular processes, such as protein synthesis and regulation of gene expression [1,2,3,4,5,6]. Missense mutations occurring in these RNA binding proteins that alter protein–RNA interactions may cause significant deviation from the normal function of these proteins, potentially leading to various disorders including cancer [7,8]. Indeed, several comprehensive studies of the structural nature of mutations in cancer and rare Mendelian diseases have shown that mutations located on binding interfaces may induce macromolecular interaction perturbations and play “driver” or “damaging” roles in many cancers and Mendelian diseases [9,10,11,12,13,14,15,16]. Quantifying the effects of missense mutations on specific protein–RNA interactions requires assessing the binding affinity changes upon introducing mutations, which can be accurately measured by traditional mutagenesis technologies and recently developed high-throughput experimental methods. However, traditional experimental methods used to measure binding affinity, such as surface plasmon resonance [17] and isothermal titration calorimetry [18], are costly and time-consuming, whereas the empirically derived binding landscapes are limited to specific systems with affinities and solubilities within the concentration windows accessible by these high-throughput methods [19,20,21,22]. Therefore, developing reliable computational approaches provides an alternative way to investigate the effects of mutations on proteins and their interactions with other molecules on a large scale, and it will facilitate the identification of functionally important missense mutations and the discovery of the molecular mechanisms that cause diseases.

Many efforts have been made to computationally predict and model the effects of missense mutations on protein stability and protein–protein interactions [23,24,25,26,27,28,29,30,31,32]. However, predicting the impacts of mutations on protein–nucleic acid interactions has been more intractable and very few computational methods have been proposed [33,34,35,36,37]. One of the reasons hindering the development of methods is due to the complexity of nucleic acid chemistry and binding, which limited the availability of high-quality experimental data. Moreover, the interactions between protein–DNA and -RNA are different, which was clarified by a detailed comparison at the atomic contact level [38]. Recently, we developed a computational approach to estimate the impacts of missense mutations on protein–DNA interactions using molecular mechanics force fields and statistical potentials [34]. Peng et al. combined modified Molecular Mechanics/Poisson-Boltzmann Surface Area (MM/PBSA)-based energy terms with additional knowledge-based terms to predict the protein–DNA binding affinity changes upon single mutations [35]. Pires et al. used graph-based signatures to model the effects of single mutations both on protein–DNA and -RNA binding [33]. The three methods mentioned above estimate the effects of mutations on the interactions by quantitatively calculating the changes in binding affinity. In addition, several classification computational approaches have been proposed for predicting the binding hot spots at protein–RNA binding interfaces [36,37,39]. Therefore, although some advancements have been made, the issue of predicting the effects of missense mutations on protein–RNA interactions is still at the initial stage.

To address this need, we introduced a new computational method, PremPRI, for characterizing the effects of missense mutations on protein–RNA interactions by calculating the binding affinity changes quantitatively. PremPRI uses a novel multiple linear regression scoring function composed of 11 sequence- and structure-based features, and achieves significantly better performance than other predictors. PremPRI can be applied to many tasks, such as finding potential disease-causing and cancer driver missense mutations and understanding their molecular mechanisms, and designing inhibitors of protein–RNA interactions. PremPRI is freely available at https://lilab.jysw.suda.edu.cn/research/PremPRI/.

## 2. Results and Discussion

### 2.1. Multiple Linear Regression Model of PremPRI

The PremPRI model is built using a multiple linear regression algorithm and composed of eight structure- and three sequence-based features. The *p*-value and importance for each feature are shown in Appendix A that indicate all of the features contribute significantly to the model. To investigate the linear associations between different features, we performed multicollinearity analysis. Appendix A presents that the variance inflation factor (VIF) of each feature is less than three, indicating low collinear relationships among 11 independent variables in the PremPRI model.

The performance of PremPRI trained and tested on the S248 set is shown in Figure 1a and Table 1. The Pearson correlation coefficient between experimental and calculated binding affinity change is 0.72 and the corresponding RMSE and slope is 0.76 kcal·mol^−1^ and 1.00, respectively.

PremPRI takes about four minutes to perform the calculation for a single mutation in a protein–RNA complex with ~330 residues and ~70 nucleotides, and requires 30 s for each additional mutation introduced in the same complex.

### 2.2. Performance on Three Types of Cross-Validation

Overfitting is one of the major problems in machine learning. In order to check whether our method has this issue, we performed three types of cross-validation (CV1, CV2, and CV3). In CV1 and CV2, we randomly chose 50% and 80% of all mutations from the S248 dataset to train the model, respectively, and used the remaining mutations for blind testing; both procedures were repeated 100 times. In CV3, the model was trained and tested using nonoverlapping sets of protein–RNA complexes. Namely, we left one complex and its mutations out as the test set and trained the model using the rest of the complexes/mutations (leave-one-complex-out validation); this process was repeated for each complex.

The correlation coefficients for 100 times cross-validation of CV1 and CV2 are shown in Figure 1b, and the R of each round is > 0.42. The average values of R and RMSE for 100 times CV1 and CV2 are 0.68 and 0.80 kcal·mol^−1^, respectively (Table 1). In the leave-one-complex-out validation (CV3), the correlation coefficient reaches 0.61 (Figure 1c) and RMSE = 0.87 kcal·mol^−1^ (Table 1). Moreover, PremPRI does not present bias to the different categories of mutations, including alanine-scanning and nonalanine-scanning mutations, interfacial and noninterfacial mutations, and the mutations from protein-single stranded RNA complexes (Protein-ssRNA) and protein-double stranded RNA complexes (Protein-dsRNA) (Appendix A). The performance for each category remains relatively high in CV3 with R of ~0.60 (Appendix A). In addition, analysis of the variation of weighting coefficient for each feature in the three types of cross-validation further illustrates that the PremPRI model does not overfit on its training set and all features contribute significantly to the energy function (Appendix A).

### 2.3. Comparison with Other Methods

We compared our method with three other available computational methods, mCSM-NA [33], PrabHot [36], and FoldX5.0 [40], developed for predicting the effects of mutations on protein–RNA interactions. mCSM-NA calculates the changes in protein–DNA/RNA binding affinity using graph-based signatures. PrabHot is a classification method and uses a combination of sequence, structure, and residue-interaction-network-based features to identify hot spots at protein–RNA binding interfaces. FoldX5.0 calculates the binding affinity changes using an empirical energy function and is parametrized on experimentally determined unfolding free energy changes. All of them are machine learning approaches and the training datasets for parameterizing mCSM-NA and PrabHot are shown in Appendix A. The number of common mutations between S248 and the training datasets of mCSM-NA and PrabHot is 16 and 92, respectively. We did not compare S248 with the training dataset of FoldX5.0 since it is composed of unfolding free energy changes.

We applied all three methods on the S248 dataset, and the correlation coefficients are 0.20 (RMSE = 1.57 kcal·mol^−1^) and 0.24 (RMSE = 5.41 kcal·mol^−1^) for FoldX and mCSM-NA, respectively (Table 2). Moreover, we performed ROC and precision-recall analyses in order to estimate the performance of different methods to predict the highly decreasing mutations ($\Delta \Delta G_{exp}$ ≥ 1 kcal·mol^−1^, the same cutoff was used in PrabHot to define the hot spots), and the results are shown in Table 2 and Figure 2a. For PremPRI, the leave-one-complex-out validation results are used. Although it is not a completely identical comparison, the significantly large differences of the values of R, RMSE, AUC-ROC, AUC-PR, and MCC between PremPRI (CV3) and the other methods can prove the better performance for our method. In addition, our method performs well for both interfacial and noninterfacial mutations (Figure 2b), and the only statistically significant correlation coefficient is observed for mCSM-NA to predict interfacial mutations.

Furthermore, we applied all methods on a test case that includes four single mutations occurring in the highly conserved triad Thr-Met-Gly of Ribosomal protein L1 [41]. All four mutations lower the protein–RNA binding largely (The experimental values of $\Delta \Delta G_{exp}$ are presented in Appendix A). Crystal structure of Ribosomal protein L1 from Thermus thermophilus (TthL1) in complex with a specific80 nt fragment of 23S rRNA (PDB ID: 3U4M) is used to perform the calculations. Our training dataset of S248 includes one mutation of T217A from this complex of 3U4M, which was excluded from the training dataset when testing on this case. The predictions shown in Appendix A indicate that the PremPRI has the best performance and predicts all four mutations as highly decreasing mutations.

Next, we would like to illustrate the main differences between PremPRI and the other predictors of mCSM-NA, PrabHot, and FoldX. The approaches of mCSM-NA and PrabHot use powerful machine learning algorithms composed of several dozens of features to evaluate the binding affinity changes and they do not provide each feature’s contribution for each mutation. In addition, mCSM-NA calculates the changes in both protein–DNA and –RNA binding affinity and its training dataset is dominated by the mutations from protein–DNA complexes (see Appendix A). PrabHot is a classification method and identifies hot spots only at protein–RNA binding interfaces. On the other hand, the FoldX and PremPRI methods use multiple linear regression algorithms with very few interpretable features, and they produce mutant structure and perform structure optimization. The molecular structures of mutants have been widely used by researchers in the fields of life sciences and human health, such as finding disease driver mutations and understanding the molecular mechanisms, and designing drugs and deciphering the mechanisms of drug-resistant mutations. In comparison with PremPRI, FoldX uses an empirical energy function that is parametrized on experimentally determined unfolding free energy changes to calculate the binding affinity changes.

Upon the limitation of the study, the PremPRI method cannot predict the impact of multiple amino acid substitutions on protein–RNA interactions. The number of multiple mutations with experimentally determined binding affinity changes is very small, which cannot be used to train an accurate model. Although the users can calculate the impact of every single mutation in multiple mutations separately using our method, we know that the effects of lots of multiple mutations are not simply the sum of the effects of the single mutations. Moreover, although our training set contains more mutations compared with other methods, it is still not big enough, which may limit the prediction accuracy of the method on different kinds of complexes and mutations. More experimentally measured values of changes in binding affinity need to be collected and included in the training dataset to further improve the method’s performance on various test cases.

### 2.4. Online Webserver

#### 2.4.1. Input

The PremPRI webserver requires the 3D structure of a protein–RNA complex that can be retrieved from the Protein Data Bank with the PDB code input or provided by a file with the atomic coordinates uploaded by the user (Figure 3 and Appendix A). In either case, the structure file must include at least two chains, one is protein and the other is RNA. After the structure is retrieved correctly, the server will display a 3D view of the complex structure and list the corresponding protein or RNA name for each chain (Appendix A). As a second step, two interaction partners should be defined, and one or multiple chains can be assigned to each partner. Only assigned chains will be considered during the prediction. As a third step of selecting mutations, three options are provided allowing users to do large-scale mutational scanning (Figure 3 and Appendix A). In the option of “Specify One or More Mutations Manually”, the user can submit the specified mutations and visualize each mutated site in the protein–RNA complex structure. “Alanine Scanning for Each Chain” option is used to perform alanine scanning for each protein chain. The “Upload Mutation List” option allows users to submit a list of mutations specified in the uploaded file.

#### 2.4.2. Output

For every single mutation in a protein–RNA complex, the PremPRI server provides: ΔΔG (kcal·mol^−1^), predicted binding affinity change, and positive and negative signs correspond to the mutations decreasing and increasing binding affinity, respectively; Interface (yes/no), shows whether the mutation occurs at the protein–RNA binding interface. When a residue’s solvent-accessible surface area in the complex is lower than in the unbound partner, it is defined as located at the interface. Furthermore, for each mutation, PremPRI provides an interactive 3D viewer which shows the noncovalent interactions between the residue in the mutated site and its adjacent residues/nucleotides, generated by Arpeggio [42]. The minimized wild-type and mutant complex structures are used to show the interactions. An example is provided in Figure 3 and Appendix A.

## 3. Methods

### 3.1. Experimental Datasets Used for Training

First, we used ProNIT and dbAMEPNI databases to compile our training dataset. ProNIT [43] includes experimentally measured values of changes in binding affinity (ΔΔG) between proteins and nucleic acids upon single amino acid substitutions along with the experimentally determined 3D structures of protein-nucleic acid complexes. The dbAMEPNI [44]—a recently proposed database—collected more experimentally measured binding affinity changes between protein and nucleic acid induced by alanine-scanning mutations compared to ProNIT. In order to construct an accurate and cleaned training dataset, we first removed the following complexes and mutations from the above databases, including the complexes with modified residues/nucleotides located at the protein–RNA binding interface, the complexes in which the protein length is less than 20 amino acids or RNA length is less than five nucleotides, and the mutations occur at metal coordination sites. Then, to avoid inconsistencies between the RNA sequences used to measure binding affinity and the sequences presented in the 3D complex structures used to develop a prediction model, we compared the sequence similarity between the sequences at the protein–RNA binding sites and the corresponding ones used to measure binding affinity. The RNA sequences in binding affinity measurements were either obtained from the ProNIT database or manually collected from the corresponding references. The entries with high sequence similarity (80%) were retained. In addition, there are three mutations with multiple experimental measurements of ΔΔG. Since the differences between maximal and minimal ΔΔG values for these cases are < 1 kcal·mol^−1^, the average value was used. As a result, 108 single mutations in 30 protein–RNA complexes from the ProNIT and dbAMEPNI databases were retained in our training dataset.

Secondly, 140 additional mutations obtained from the PrabHot benchmark and independent test sets were added into our training dataset [36], and they satisfied all the above criteria. PrabHot is a classification computational method for identifying hot spots at the protein–RNA binding interfaces. Therefore, the final experimental dataset used to parameterize our PremPRI model includes 248 single mutations from 50 protein–RNA complexes (named as S248) (Appendix A). The number of mutations for each protein–RNA complex is shown in Appendix A. In the S248 dataset, only 16 mutations have experimental pH values. Thus, the neutral pH was chosen at which the default charged states were assigned to the ionizable residues. We also compared our training dataset of S248 with them used for developing the mCSM-NA [33] and PrabHot methods [36], and the details are shown in Appendix A.

### 3.2. Structural Optimization Protocol

Three-dimensional structures of protein–RNA complexes were obtained from the Protein Data Bank (PDB) [45]. The biological assembly 1 of crystal structure or the first model of nuclear magnetic resonance (NMR) structure was used as the initial wild-type structure. Next, we used the BuildModel module of the FoldX software package [24,40] to produce mutant structures. Then, the VMD program [46] was applied to add missing heavy side-chain and hydrogen atoms to both wild-type and mutant structures using the topology file of the CHARMM36 force field [47]. After that, we carried out a 1000-step energy minimization for each complex in the gas phase during which the harmonic restraints with a force constant of 5 kcal·mol^−1^·Å^−2^ were applied on the backbone atoms of all residues. The NAMD program v2.12 [48] and the CHARMM36 force field [47] were used to perform the energy minimization. A 12 Å cutoff distance for nonbonded interactions was applied to the molecular systems and lengths of hydrogen-containing bonds were constrained by the SHAKE algorithm [49]. The flowchart of the structural optimization protocol is shown in Appendix A. The minimized wild-type and mutant protein–RNA complexes were used for the following calculations of energy features.

### 3.3. The PremPRI Model

The multiple linear regression scoring function of PremPRI is composed of 11 features and each of them has a significant contribution to the quality of the model (*p*-value < 0.01, *t*-test). The *p*-value and the importance of each feature are presented in Appendix A and the description is illustrated below:
$\Delta \Delta E_{vdw}$ is the difference of van der Waals interaction energies between mutant and wild type ($\Delta \Delta E_{vdw}=\Delta E_{vdw}^{mut}-\Delta E_{vdw}^{wt}$). $\Delta E_{vdw}$ is the difference of van der Waals energies between a protein–RNA complex and each binding partner (Partner 1: protein; Partner 2: RNA), which is calculated using the ENERGY module of the CHARMM program [50].$\Delta \Delta E_{vdw.re}$ is the difference of van der Waals repulsive energies between mutant and wild type. Here, the van der Waals repulsive energy only counts the repulsion between the residue at the mutated site and the nucleotides.$\Delta \Delta E_{elec}$ is the difference of electrostatic interaction energies between mutant and wild type ($\Delta \Delta E_{elec}$ = $\Delta E_{elec}^{mut}-\Delta E_{elec}^{wt}).\;\Delta E_{elec}$ is the electrostatic interaction energy between the residue at the mutated site and its contact residues/nucleotides. If any side-chain atom/base of a residue/nucleotide is located within 10 Å from any side-chain atom of the mutated site, we defined it as a contact residue/nucleotide. The calculation is carried out using the ENERGY module of the CHARMM program.$N_{inter}$ is the number of amino acids at the protein–RNA binding interface. If the solvent-accessible surface area of a residue in the protein is more than that in the complex, we define it as the interface residue. The SASA module of CHARMM is used to calculate the solvent-accessible surface area.$R_{L/SA}$ is the ratio of protein length and its surface area. $R_{L/SA}=\frac{\mathrm{Length}}{\mathrm{SASA}}\;,$
Length and SASA is the total number of residues and the solvent-accessible surface area of unbound protein, respectively. The structure of the unbound protein is extracted from the minimized wild-type complex structure.Closeness of the node of the mutated site in the residue interaction network. It is defined as:(1)$$C\left(u\right)=\left(n-1\right)/\underset{v\in V,\;v\neq u}{\sum }d\left(u,\;v\right)$$
where d(u, v) is the shortest-path distance between the node u of the mutated site and any node v. V is the set of all nodes and n is the number of nodes in the residue interaction network. The shortest-path distance between two nodes refers to the minimum number of nodes that reach from one node to the other [51]. The Cα atom of a residue is considered as a node. If the distance between two Cα atoms is <6 Å, we define them as having a direct interaction. Closeness is calculated using the Python package NetworkX [52].ΔSA is the difference of solvent accessible surface areas between mutant and wild type (ΔSA = $SA_{site}^{mut}-SA_{site}^{wt})$. $SA_{site}$ is the solvent accessible surface area of the residue at the mutated site in the unbound protein that is extracted from the minimized complex structure.$P_{coil}=\frac{N_{e.c}}{N_{All}}$, $N_{e.c}$ and $N_{All}$ are the number of exposed residues in the coil conformation and all residues in the mutated protein chain, respectively. Secondary structure elements other than α-helices and β-strands are defined as coil, which are assigned by the DSSP program [53]. If the ratio of the solvent-accessible surface area of a residue in the complex and in solvent is more than 0.25 [54], we defined it as the exposed residue.ΔOMH is the difference of hydrophobicity scale between mutant and wild-type residue types. The hydrophobicity scale (OMH) for each type of amino acid was derived by considering the observed frequency of amino acid replacements among thousands of related structures, which was taken from the study of [55] directly.$\Delta P_{FWY}$ and $\Delta P_{KR-DE}:$
$\Delta P_{FWY}$ = $P_{FWY}^{mut}-P_{FWY}^{wt}$, $\Delta P_{KR-DE}$ = $P_{KR-DE}^{mut}-P_{KR-DE}^{wt}$, $P_{FWY}=\frac{N_{FWY}}{N_{All}}$ and $P_{KR-DE}=\frac{N_{KR}-N_{DE}}{N_{All}}$. $N_{FWY}$, $N_{KR}$, $N_{DE}$ and $N_{All}$ are the number of aromatic (F, W and Y), positively charged (K and R), negatively charged (D and E) and all amino acids in the mutated protein chain, respectively.

In addition, we also tested the performance of several other popular machine learning algorithms, including Random Forest (RF), Back Propagation Neural Network (BPNN), Support Vector Machine (SVM) and eXtreme Gradient Boosting (XGBoost), and the results shown in Appendix A indicate that the multiple linear regression algorithm performs the best.

### 3.4. Statistical Analysis and Evaluation of Performance

Pearson correlation coefficient (R) and root mean square error (RMSE) are used to measure the agreement between experimentally-determined and predicted values of binding affinity change. All correlation coefficients presented in the paper are significantly different from zero (*p*-value < 0.01, *t*-test). RMSE (kcal·mol^−1^) is the standard deviation of the residuals (prediction errors). The Hittner2003 test [56,57] is used for comparing whether the difference in correlation coefficients between PremPRI and other methods is significant. The changes in the area under the receiver operating characteristics curve (AUC of ROC) are tested by the Delong test [58]. All tests are implemented in R.

Receiver Operating Characteristics and precision-recall analyses were performed to quantify the performance of different methods in distinguishing mutations highly decreasing binding affinity ($\Delta \Delta G_{exp}$ ≥ 1 kcal·mol^−1^) from others. True positive rate (TPR) and false-positive rate (FPR) is defined as TPR=TP/(TP + FN) and FPR=FP/(FP+TN) (TP: true positive; TN: true negative; FP: false-positive; FN: false negative), respectively. In addition, the maximal Matthews correlation coefficient (MCC) value is reported for each method by calculating the MCC across a range of thresholds:(2)$$\mathrm{MCC}=\frac{TP*TN-\mathrm{FP}*\mathrm{FN}}{\sqrt{\left(TP+FP\right)\left(TP+FN\right)\left(TN+FP\right)\left(TN+FN\right)}}$$

## Figures and Tables

**Figure 1 ijms-21-05560-f001:**
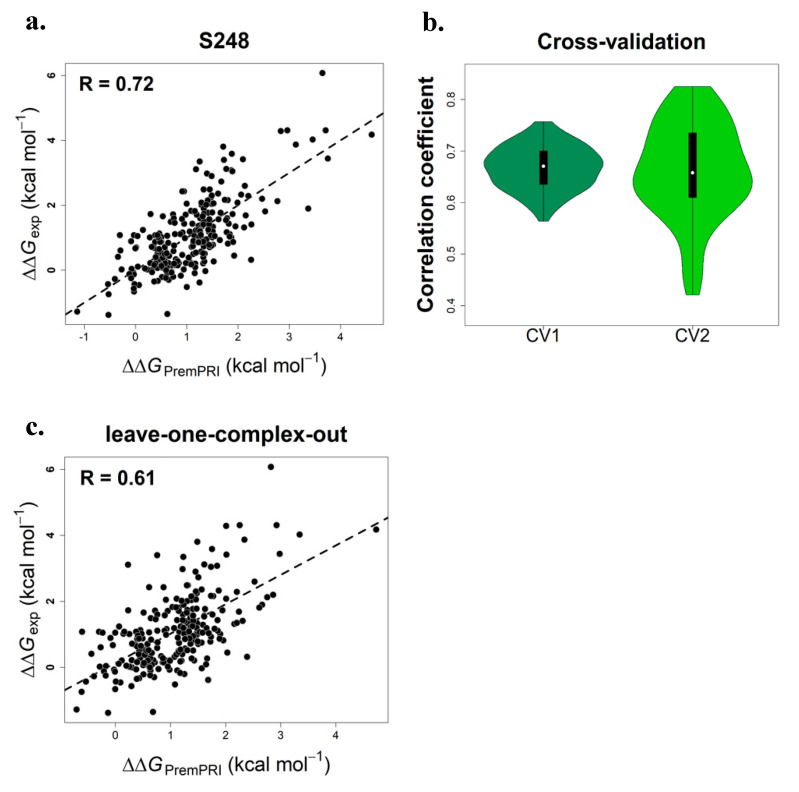
Pearson correlation coefficients between experimental and calculated changes in binding affinity for PremPRI trained and tested on S248 dataset (**a**), using two types of cross-validation (CV1 and CV2) (**b**) and performing leave-one-complex-out validation (CV3) (**c**), respectively.

**Figure 2 ijms-21-05560-f002:**
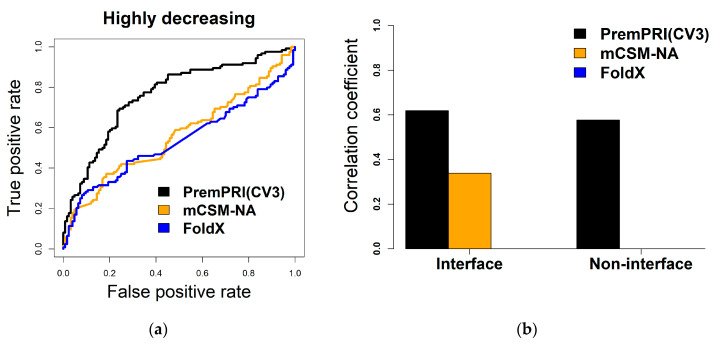
Performance of three methods of PremPRI, mCSM-NA, and FoldX applied to the S248 dataset. The leave-one-complex-out validation (CV3) results of PremPRI are used. (**a**) ROC curves for predicting highly decreasing mutations. The number of highly decreasing mutations ($\Delta \Delta G_{exp}$ ≥ 1 kcal·mol^−1^) in S248 is 124. (**b**) Pearson correlation coefficients between predicted and experimental $\Delta \Delta_{G}$ for mutations located at protein–RNA binding interface and noninterface. Only correlation coefficients that are significantly different from zero are shown in the figure (*p*-value < 0.01, *t*-test).

**Figure 3 ijms-21-05560-f003:**
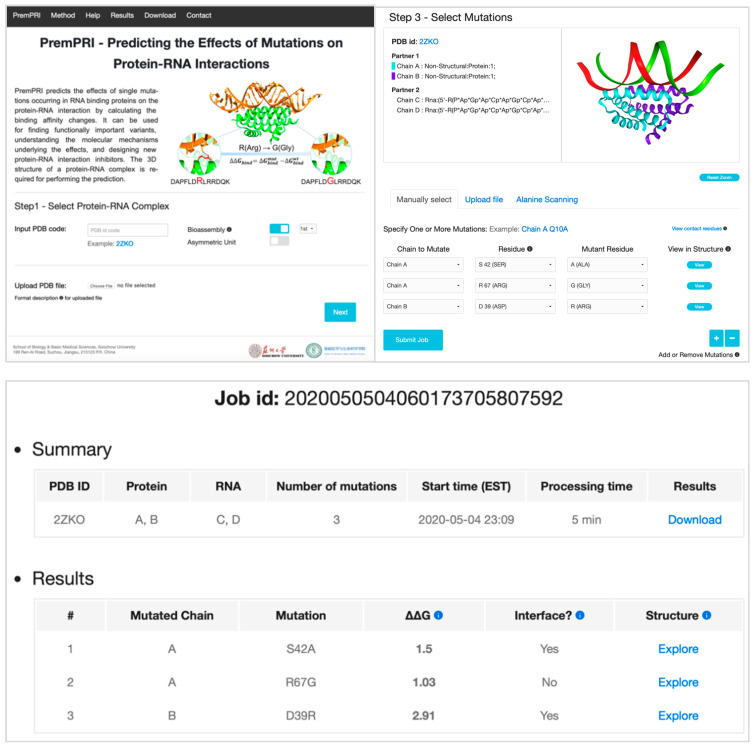
Left corner: the entry page of PremPRI server; right corner: the third step for selecting mutations and three options are provided: “Specify One or More Mutations Manually”, “Upload Mutation List” and “Alanine Scanning for Each Chain”, see also Appendix A; and bottom: final results, see also Appendix A. “Processing time” refers to the running time of a job without counting the waiting time in the queue.

**Table 1 ijms-21-05560-t001:** The performance for PremPRI trained and tested on the S248 dataset and performing three types of cross-validation.

Method	R	RMSE	Slope
**PremPRI**	0.72	0.76	1.00
**PremPRI (CV1)**	0.68	0.80	0.94
**PremPRI (CV2)**	0.68	0.80	0.95
**PremPRI (CV3)**	0.61	0.87	0.89

R: Pearson correlation coefficient between experimental and predicted ΔΔG values. RMSE (kcal·mol^−1^): root-mean-square error. Slope: the slope of the regression line between experimental and predicted ΔΔG values. All correlation coefficients are statistically significantly different from zero (*p*-value < 0.01).

**Table 2 ijms-21-05560-t002:** Comparison of methods’ performances on the S248 dataset.

Method	R	RMSE	AUC-ROC	AUC-PR	MCC
**PremPRI (CV3)**	0.61	0.87	0.76	0.76	0.45
**mCSM-NA**	0.24 *	5.41	0.56 *	0.60	0.22
**FoldX**	0.20 *	1.57	0.53 *	0.59	0.24
**PrabHot**	-	-	0.58 *	0.61	0.26

R: Pearson correlation coefficient. RMSE (kcal··mol^−1^): root-mean-square error. AUC-ROC: the AUC values of ROC curves. AUC-PR: the AUC values of precision-recall curves. MCC: maximal Matthews correlation coefficient. All correlation coefficients are statistically significantly different from zero (*p*-value < 0.01). * indicates a statistically significant difference between PremPRI and other methods in terms of R and AUC-ROC with *p*-value < 0.01 (Hittner2003 and DeLong tests are used for comparing correlation coefficients and AUC values, respectively).

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
