# Peer review of "PremPRI: Predicting the Effects of Missense Mutations on Protein–RNA Interactions"

_ijms, 2020, doi:10.3390/ijms21155560_

Round 1
Reviewer 1 Report
The nature of a protein-RNA binding equilibrium is quantified by the free energy change upon the binding (ΔG). The manuscript presents a new computational method of predicting the change in the ΔG due to the single amino acid mutations of the RNA binding proteins (ΔΔG). The method, named PremPRI, employs a scoring function which is composed of 11 structure- and sequence-based components in which the weighting coefficients are multiplied. The coefficients are determined using the multiple linear regression algorithm on the basis of a set of 248 experimental ΔΔG, so that the calculated ΔΔG is fitted to the experimental one. The authors demonstrate that the performance of the PremPRI is far superior to the three methods currently available (although two of the three methods are not specialized for the protein-RNA binding). All the computations, analyses, and the statistical justifications have been done carefully. The writing and argument are clear. The reviewer thus recommends the publication of the manuscript after the following minor points are addressed.
1) On page 5, line 198-190. The authors state “The PremPRI model is ~ and composed of ten structure- and one sequence-based features”. However, the two components, ΔPFWY and ΔPKR−DE, are likely to be the sequence-based features. In addition, a hydrophobicity scale (OMH, on page 4, line 154-157) seem to be a feature of each single amino acid irrespective of whether it is exposed to water or not in a protein (i.e., it is independent from a protein structure), although the OMH is parametrized using the structural information of many proteins (it is stated in Table 1 of Ref. 49 as “These were derived, respectively, from vapor pressures of side-chain analogues and counts of buried and exposed residues in globular proteins”.)
2) On page 3, line 134-136. The reviewer is unfamiliar with RL/SA although it is clearly defined. The authors should present whether it is their original idea or not.
3) On page 8, line 273-275. The ΔΔG of four single mutations at three position (Thr-Met-Gly of Ribosomal protein L1) are presented in Table S7.
Reviewer 2 Report
Missense mutations can disturb or impact the RNA-protein interactions critical for many cellular processes. Experimental biologists often struggle with measuring the impact of point mutations in RNA binding proteins on the interaction with RNA. Hence, a good computational tool for this would be a great development. The manuscript by Zhang et al. presents a new computational method PremPRI together with an online web server that quantitatively calculates the effect of single aa mutations on the protein-RNA binding affinity. The PremPri was parameterized using the large experimental dataset and the Authors compared their tool with 3 other computational tools for the prediction of the effect of mutations on the protein-RNA binding. Presented results showed a good agreement between experimental and predicted binding affinity values and support better PremPri performance than other tested tools.
The PremPRI is limited to RNA-protein complexes with the experimentally confirmed 3D structure but should interest researchers who study how point mutations impact the function of proteins.
I have only two minor comments for the Authors
L 35-37: The Authors have motivated the manuscript by claiming that experimental methods such as SPR and ITC are costly and time-consuming. However, many simpler and cheaper methods are available and should be mentioned here.
Supplementary data: It was difficult to analyze the data presented in Tables S2, S4, and S6. These tables should be corrected.
Reviewer 3 Report
In this manuscript, Zhang et al. developed PremPRI, a novel tool for predicting the influence of a single mutation in a protein sequence on protein-RNA interactions. The authors showed that PremPRI had better performances than existing software. In addition, authors developed a web server for PremPRI to make it easier for other researchers to use. While this article is well written and the developed tool is useful to the community, the manuscripts should require some revisions.
Comments
- Although the authors described many good results for PremPRI in the manuscript, the authors provide little discussion of the results. I think there is much to discuss in this paper, for example, 1. Why did PremPRI outperform other software?, 2. Is it possible to extend the software to consider not only single mutations but also multiple mutations?, and 3. What are the limitations and the future perspectives of this software?. The authors should create a Discussion section in this paper and carefully discuss the results.
- In the present title, readers cannot understand whether “single mutations” in the title means mutations in protein sequences or RNA sequences. Authors should specify that “single mutations” means mutations in protein sequences in the title.
- I could not understand the information in Fig. S1. The authors should provide the information by a table rather than the pie chart.
- Because Fig. S3 and S4, which are figures about the PremPRI web server, are important results in this paper, these figures should be in the main text, not in the Supplementary Materials.
Round 2
Reviewer 3 Report
My initial comments have been appropriately addressed in the reply, and I find that authors have improved the paper in a significant way. As the methodology proposed in this manuscript will be useful to the community, I recommend that this manuscript be accepted for publication.